# Potential Owner-Related Risk Factors That May Contribute to Obesity in Companion Dogs in Aotearoa New Zealand

**DOI:** 10.3390/ani12030267

**Published:** 2022-01-21

**Authors:** Rachel Forrest, Leena Awawdeh, Fiona Esam, Maria Pearson, Natalie Waran

**Affiliations:** 1Eastern Institute of Technology, Hawke’s Bay, 501 Gloucester Street, Taradale, Napier 4112, New Zealand; mpearson@eit.ac.nz (M.P.); nwaran@eit.ac.nz (N.W.); 2Companion Animals New Zealand, Wellington 6141, New Zealand; fiona.esam@gamil.com

**Keywords:** Aotearoa, canine, companion animal, diet, dog, dog–owner relationship, human behaviour, New Zealand, obesity, pet

## Abstract

**Simple Summary:**

Even though dogs are an integral part of many Aotearoa New Zealand (NZ) households, there is a shortage of studies investigating the NZ dog owners’ attitudes towards their responsibilities. In addition, very little is known about factors contributing to dogs being overweight or obese in NZ. Understating the factors that influence human behaviour toward their pets is crucial to improving animal welfare. The current study aims to identify potential owner-related risk factors contributing to canine obesity in NZ. Identifying these potential risk factors will help inform intervention to maintain a healthy weight in companion dogs and ensure NZ dogs live a good life.

**Abstract:**

Approximately a third of all Aotearoa New Zealand (NZ) households include a dog, with 28% of these dogs being overweight or obese, conditions that are associated with many serious health issues. Therefore, healthy weight interventions that focus on the owner’s role are of great importance to companion animal welfare in NZ. Accordingly, the present study explores the feeding practices associated with NZ dogs and identifies potential owner-related risk factors contributing to these animals being overweight or obese. The current study used data collected from a survey conducted online in 2019 between January and March of NZ residents over 18. Along with demographic questions, the respondents were asked questions regarding their dog’s body condition and diet questions related to the body. Nearly a quarter (26%, *n* = 609) of the survey participants (*n* = 2358) owned at least one dog. The current study reported that increasing age range, household income and the number of children increased the likelihood of having a dog while increasing qualification level and living in a town/city decreased the likelihood. The majority of the respondents fed their dog(s) treats (59%) and 85% fed them specialised food bought from a pet shop, veterinary clinic and/or different online sources. Just over a third of the participant (39%) reported that they fed their dog(s) biscuits from the supermarket, 36% fed their dog(s) raw meat, and 34% of respondents fee their dog(s) table scraps/human food. These results suggest that many dog owners feed their dog(s) various food types, making it a challenging task to determine the exact amount required from each type in order not to exceed caloric intake. Disagreement regarding the correct body condition were reported among twenty per cent of the respondents. This finding indicates a knowledge gap among the NZ dog-owning population that may negatively affect their dogs’ welfare and wellbeing. Future research into pro-equity approaches to address these issues is needed so that dogs in NZ can live not only a good life but also their best life.

## 1. Introduction

The second most popular companion animal in Aotearoa New Zealand (NZ) is dogs, with approximately 34% of households owning at least one pet dog [1]. It has long been acknowledged that dog ownership is positively influenced by demographic variables such as a non-urban location, increased size of the house or property, number of children in the family, education level, and household income [2,3]. In April 2020, only 26% of Wellington households included a dog, while 37% was reported for urban/city households overall, the latter being a substantial increase from 27% in 2015 [1]. Not surprisingly, more rural households (44%) had dogs compared to non-rural areas [1]. Higher rates of dog ownership were also observed in households with adults under the age of 25 years (47%), children (42%), earning a higher income (37%), and that were Māori (46%), whereas only 29% of Pacifica households had a dog [1]. Māori are the indigenous people of NZ.

In NZ, households with dogs have increased from 29% in 2011 and 28% in 2015 to 34% in 2020. This recent increase may have been at least in part due to the COVID-19 pandemic [4], with pet ownership being a protective factor against depression, anxiety and stress [5]. With the increase in the dog ownership rate, the overall health and wellbeing of owners and pets have to be optimised [6,7,8]. Furthermore, 78% of NZ dog owners consider their canine companions family members [1]. As more owners consider their pets an integral part of the family, their pet’s health status and addressing health problems such as obesity have become increasingly important, impacting both owner and pet wellbeing [9,10,11,12]. Unfortunately, obesity is common in dogs (along with their human counterparts) and is associated with many serious health conditions that can reduce life expectancy. For example, excess weight can predispose dogs to develop diseases such as diabetes mellitus, osteoarthritis, and urinary incontinence [13]. Globally, studies have estimated that canine obesity prevalence ranges from 10% to 40% [10,11,14,15], a trend that appears to be increasing [13]. Gates et al. [16] reported that 28% of dogs were overweight/obese in the North Island of NZ.

Like humans, dogs’ excess weight is a complex health challenge, especially when animals exceed their optimal weight by more than 10–25% [17]. Although a similar obesity pattern has been reported in human and pet populations, the factors contributing to these associations have not been characterised [18,19,20]. A shared lifestyle can mean that those factors contributing to human obesity may also impact negatively on their pets [20]. Incorrect perceptions of healthy body weight can promote both human and pet obesity; if a weight problem is not acknowledged, it is unlikely to be resolved. Dog owners need to be able to recognise when their pet is gaining excess weight and take suitable measures as pet dogs are usually entirely dependent on their owner for their diet requirements. Some studies demonstrate poor owner body condition perception in recognising obesity in dogs [21,22].

Even though dogs are an integral part of many NZ households, there is a dearth of research investigating NZ dog owners’ attitudes towards their responsibilities. In addition, very little is known about factors contributing to dogs’ overweight/obesity in NZ. The current Furry Whānau Wellbeing research project was funded by the New Zealand Companion Animal Trust (NZCAT) [23] to explore the perspectives of New Zealanders towards their pets’ (dogs and cats) health and wellbeing and the factors that may influence their attitudes. In this study, NZCAT research project data relating to dogs’ diets and the perceptions of body condition were explored to identify possible owner-related risk factors that may be associated with the development of canine obesity. Obesity in dogs negatively impacts their physical health and reduces their overall quality of life. In order to have a good life, dogs need to be able to “express a rich behavioural repertoire, use their abilities, and fulfil their potential through active engagement with their environment” [24]. We hypothesise that the behaviours and perceptions of some NZ dog owners may reveal potential canine obesogenic risk factors. The current study’s findings will help direct future investigations and interventions to maintain companion dogs’ healthy weight and ensure NZ dogs live a good life.

## 2. Materials and Methods

The NZCAT Furry Whānau Wellbeing research was carried out with approval from the Research and Ethics Committee at the Eastern Institute of Technology (EIT) (REAC ref 19/53). A full description of all aspects of the research can be found in the previous report [23].

### 2.1. Participants and Data Collection

The authors developed the survey (the EIT Pet Survey 2019) in consultation with topic experts and local Māori representatives and made it available online to New Zealanders via Survey Monkey© for three months in 2019 between 8 January to 31 March [23,25]. The survey was promoted through various social media (for example, Facebook, Instagram, LinkedIn) and via email using a snowballing approach. Participants were not paid and IP addresses were not controlled as it was expected that some households might share devices. For convenience, questions related to demographics, dog diet, and body condition that were part of the much larger EIT Pet Survey 2019 can be found in Appendix A.

### 2.2. Statistical Analysis

The approach used for the data analyses is described in detail in Forrest et al. [23,25]. In summary, a general inductive approach was used to analyse the qualitative collected data from the diet-specific questions [26]. Percentages were used to describe the responses to survey questions using SPSS Statistics (version 25). Forward stepwise binary regressions were used to explore whether demographic and/or household factors (as described in Section 2.1) were associated with what an owner feed their dog(s). Each food type was assessed individually, selecting a binary variable (yes = 1, no = 0). The association of demographic and household factors on the answer choices for each of the health/care statements were explored using cross-tabulations along with Chi-square and *z*-tests (α = 0.05). Any potential owner-related canine obesity risk factors identified were compared to the demographic factors associated with owning a dog.

## 3. Results

### 3.1. Demographic Description of the Respondents

Nationally, the EIT Pet Survey 2019 was completed by 2744 people. A detailed description of these respondents is provided in the *Furry Whānau Wellbeing* report [23]. In summary, almost all of the respondents (92.3%) were female, 83.4% selected NZ European as their ethnicity and 8.3% identified as Māori. Each age bracket was represented for females and males and Māori and non-Māori. Each income bracket and qualification level was represented, with similar gender and Māori and NZ European percentages. A quarter (25%) of the respondents had had a rural upbringing, while 68% grew up in a town or city. At the time of the survey, approximately 76% of the respondents lived in a town or city. Ethnicity and gender were not associated with the percentage of respondents who had a rural upbringing or were currently living in a town/city. 

### 3.2. Dog Companionship (Ownership)

The pet companionship survey questions were answered by 2358 respondents (current pet ownership was not required to respond to the survey). Of these, 37.5% (*n* = 885) of respondents owned both cats and dogs, while 26% (*n* = 609) owned dogs only. Cat-only owners represented 28% (*n* = 652) of respondents, while 9% (*n* = 212) of the respondents did not own a dog or cat at the time of completing the survey. Of all respondents, the demographic information from 1842 was available for use in the binary regression analysis for dog ownership (yes, no). Age range (odds ratio 1.128), household income (odds ratio 1.243), qualification level (odds ratio 0.917), town-living (odds ratio 0.442), and the number of children (odds ratio 1.267) were found to be (*p* < 0.001) associated with dog ownership. Increasing age range, household income, and number of children increased the likelihood (odds) of having a dog while increasing qualification level and living in a town/city decreased the likelihood. The residence location was linked to the number of dogs (*p* < 0.001), with those living in a town or city owning fewer canine companions. No other factor significantly impacted the number of pet dogs. 

### 3.3. Demographic Characteristics of Dog-Owning Respondents

Not all the dog-owning respondents disclosed all of their demographic information, thus the total sample number the percentages are calculated from varies and is indicated in brackets. As expected, the majority of the dog-owning respondents were female (93.6%; with 6.4% male and 0.3% gender diverse, total sample *n* = 1520) and NZ European (79.1%; with 9.4% Māori, total sample *n* = 1495). The representation from each age bracket ranged from 8.5% to 24.3% (8.5% > 64 years, 17.4% 55–64 years, 24.3% 45–54 years, 20.8% 35–44 years, 18.3% 25–34 years, 10.8% 18–24 years, total sample *n* = 1520). The representation for each income bracket ranged from 3.8% to 31.1% (3.8% < 14 K NZD, 20.4% 14–48 K NZD, 20.9% 48–70 K NZD, 23.8% 70–100 K NZD, and 31.1% > 100 K NZD; total sample *n* = 1277). Of the respondent’s households, 19.4% had one adult, 56.3% had two adults, 14.3% had three, and the remainder had four or more adults (total sample *n* = 1507). Meanwhile, 69.5% did not have any children, 13.0% had one child, 13.0% had two, and the remainder had three or more children (total sample *n* = 1505). Of the dog-owning respondents, 29.6% (*n* = 449/1514) grew up rurally. At the time of the survey, 70.4% of those dog-owning respondents (*n* = 1066/1514) lived in a town or city.

### 3.4. Dog Feeding Practices

Table 1 presents the percentage of respondents that selected each of the diet choices for their dog(s) with the majority of the respondents reporting that they provide specialised dog food (58%) and treats (59%). Table 1 also presents the results from the binary logistic regression analyses indicating what factors and variables were associated with diet choice. A dog was more likely to be fed treats if the owner was female (59% versus 48% and 50% for male and gender diverse, respectively) or living in a town or city (61% versus 54% for those not living in a town or city). The likelihood of being fed treats decreased as with older owners and more children in the household. Household income, qualification level and the number of children were associated with whether or not a dog was fed a specialised diet, supermarket biscuits, or dog roll. Increasing household income was associated with a specialised diet, while the likelihood of supermarket foods (biscuits, dog roll, and wet food) and home-cooked food increased as household income decreased. A positive association was reported between responders with a higher level of qualifications and their dog(s) being fed biscuits (specialised and supermarket) and a decrease in the likelihood of a dog being fed dog roll. An increase in the number of household children was associated with a decreased likelihood of specialised food or home-cooked food and an increase in the likelihood of biscuits or dog rolls from the supermarket and being fed table scraps. The latter was also associated with an increasing number of household adults. Dogs were also more likely to be fed dog roll if their owners were Māori (42% versus 27% for NZ European and 17% Other), did not live in a city or town (33% versus 25%), or were of increased age. Dogs were more likely to be fed raw meat if their owners had a rural upbringing (41% versus 34%) and did not live in a town (43% versus 32%). As the number of adults increased, the likelihood of being fed raw meat decreased.

Of the 1499 respondents that chose to answer the dog diet question, 179 provided further comments to clarify their answer selection. A thematic analysis of these comments with representative quotes is provided in Table 2. Those who commented on the use of treats highlighted that they were either occasional or rare and/or that they were used specifically for training. Concerning specialised food, the comments acknowledged its use for health reasons and that it was bought from a variety of sources, including from national and international online suppliers with home delivery, from breeders, and from pet, veterinary and farming/rural specialist stores. Several comments also highlighted that specialised food was combined with other types of food. This was a theme that also emerged in the comments about supermarket dog biscuits, raw meat, table scraps/human food, dog roll, wet/canned food, and home-cooked food, which is evidenced in the following quote, “Dog food from vet or farm store, occasionally dog roll, and meat that is been frozen or cooked, plus additional supplements if and when required”. Several of those who chose to comment drew the distinction between raw meat and raw food diets.

There was a spectrum of comments concerning table scraps/human food, often used in combination with other food types as a teaser or appetiser. Comments about home-cooked pet food contributed to the main themes of regular diet and health reasons. Other themes that emerged from the comments suggested that few owners feed their dogs specialised food due to health requirements. Some owners had their dogs on different diets due to specific dietary-related health needs. Some respondents reported that their dogs ate cat food, and several mentioned the use of supplements and dental chews. The impact of budget was also mentioned by two respondents.

### 3.5. Body Condition and Specialised Pet Food

All of the EIT Pet Survey 2019 respondents were asked to what extent they agreed or disagreed with two statements; one describing appropriate canine body condition and the other stating the need for feeding a specialised pet diet from a pet shop or veterinary clinic. Over three quarters (78.6%, Table 3) of the respondents were in agreeance with the body condition statement, whereas only 27.6% were in agreeance with the specialised diet statement. For the statement “Dogs should have a specialised diet from a pet shop or vet clinic”, a lower percentage of those respondents with children when compared to those without children in their household either strongly agreed (6% versus 9%) or agreed (15% versus 21%), and a higher percentage disagreed (20% versus 11%).

For the statement “Dogs should have ribs, hips, and a spine that are not visible but are easily felt”, a lower percentage of Māori strongly agreed (20% versus NZ European 33% and Other 41%) and a higher percentage selected disagree (8% versus 3% for both NZ European and Other). Although not significant in the z-test, a higher percentage of Māori also selected neutral (21% versus NZ European 16% and Other 13%) for this statement. As with the other questions, there was an opportunity for the respondents to leave comments. The themes that emerged from the 133 comments provided about body condition and the use of a specialised diet from a pet or vet store are presented in Table 4 with representative quotes.

There appeared to be mixed views with regard to the body condition statement for dogs. Some respondents acknowledged that they did not know what a healthy dog should look or feel like, for example, “Not sure about the spine, rib feeling, do not want dogs overweight but not underweight either”. Some found the question difficult to answer as they considered breed variation, for example, “Animals should be of a good condition although some breeds are going to be more bony so the second to the last question is difficult to answer” and “Confused about the ribs, hip, spine question as some breeds can be healthy and show ribs. E.g., Italian greyhounds”. Breed differences were highlighted in many of the respondents’ comments, for example, “Re the hips and ribs question, the breed of the dog will depend on if you’re meant to be able to feel them. I.e., you can feel them on a boxer but you cannot on an English bulldog” and “in some leaner breeds such as heading dogs it is normal for them (when working hard) to have the last ~2 ribs just visible.—dogs should be in ideal body condition for their breed”. For dogs, age was also mentioned as a factor contributing to an animal’s body condition, “Our dog is 14 now so his spine and his are becoming more prominent with age”. Interestingly, some respondents preferred their pets with a little more fat, for example, “I do not think dogs’ ribs should be easily felt and could do with a little more weight on them than vets recommend but they should not be fat either”.

The statement from the survey about the need for a specialised diet from a vet or pet shop was met with varied responses and several respondents highlighted for dogs that “Specialised diets are not always necessary—not all supermarket brands are terrible”, that “Diet depends on needs of the dog but should be of adequate quality. Does not have to be a specialised diet from vet but must be suitable for age/breed/type/behaviour and all other factors”, and that “Specialized diet should be provided if needed for health reasons etc”. Many comments suggested that while “not all supermarket brands are terrible”, there was a need for better pet food standards and that “supermarkets should stop selling awful pet food and the standard should be raised for what is acceptable”. One respondent commented “I think there should be clearer labelling of animal food, so people actually know what’s in it and the processes/sustainability and ethics”. The perception that pet food standards were varied also held true for vet and pet stores. This is emphasised in the following comments, “Not all pet store foods are equal, and some supermarket options are good depending on the individual dog”, “Pet shop food is not always best available”, and “Not a fan of many of the vet foods as they often contain little meat and are very overpriced”. The cost of specialised foods was also highlighted as a factor with the following comment capturing the sentiment of those that commented: “Many pet owners cannot afford specialised diets for their pets, and many pets do not need them”. Several respondents went further in suggesting food alternatives, for example, “I do not think they need to be on a specialised diet from the vet but do think good quality pet food should be fed, including raw diets, over homemade meals.” In general, the comments about the two healthcare statements were focused on an individual dog’s body condition and diet requirements for good health as opposed to meeting the criteria of a generic body condition statement or where food was obtained from.

## 4. Discussion

Dog-specific factors such as breed, age and gender are all major predisposing factors for obesity. However, this condition has mostly been attributed to human-specific factors such as feeding protocols, exercise, and owner attitudes/awareness [11,27,28,29,30]. Our study aimed to interrogate the applicable data from national surveys performed recently and funded by NZCAT [23] to identify potential human-specific risk factors that contribute to companion dogs in NZ becoming overweight or obese. Based on the same survey data, potential owner-related risk factors for cat obesity have recently been reported [25]. However, caution must be used when generalising our findings to the broader population as the collective respondents’ demographics were not representative of the national population, with females being overrepresented (93.6% versus 51% in the general population) and Māori being underrepresented (9.4% versus 16.5%) [31]. The overrepresentation of female respondents was expected as it is typical of online surveys [32,33]. In NZ it has been reported that the head female usually takes on the responsibility of caring for pets in the household [34], and similar findings have also been documented in Italy and Australia [22,35]. Therefore, gender bias was not considered a major limiting factor in this study. In contrast, the under-representation of Māori is limiting. In NZ, there is inequitable access to the internet, with Māori being among those groups of people reported to have relatively low internet access [36], which may explain the underrepresentation. However, Māori households are more likely to have a dog [1] and therefore it is important that further research with an indigenous perspective is undertaken. Finally, as with any volunteer survey, there is always the potential for self-selection bias with individuals that have stronger opinions about the subject being more likely to respond [37]. Despite these limitations, the findings of this study will help inform the development of a healthy weight intervention for dogs in NZ.

### 4.1. Dog Companionship

At the time of completing the study, most participants owned at least one cat or dog (91%). Among the survey respondents, dog ownership was associated with increasing age, household income, and the number of children, while in contrast increasing qualification level decreased the likelihood of a dog sharing the household. As expected, fewer dogs were associated with owners who lived in towns or cities. These findings are consistent with another recent online survey conducted in NZ [16]. Gates et al. [38] also reported that having more than one pet was significantly associated with owner income, having children, and living rurally. The NZ studies are also consistent with overseas research, which highlight having children as a strong reason for owning a dog [2,39,40]. With regard to increasing qualification level and living in a town/city decreasing the likelihood of owning a dog, similar results have previously been reported in the United Kingdom [39], possibly reflecting lifestyle constraints (for example, lack of time and space) that make owning a dog prohibitive. Not surprisingly, place of residence was associated with the number of household dogs. Murrey et al. [39] identified that as outdoor area increases, so does the likelihood of having a dog in the household. Several factors associated with dogs being present in a household, such as being Māori or having children in the household, were also associated with incorrect perceptions of ideal body condition and feeding protocols that are potential risk factors for canine obesity.

### 4.2. Perceptions of Appropriate Body Condition

This study reports a mixed view regarding the body condition statement that dogs “Should have ribs, hips, and a spine that are not visible but are easily felt”. Most of the comments are associated with breed and age differences. In our study, approximately one in five respondents were not in agreeance with the body condition statement associated with a healthy weight in pet dogs, with a lower percentage of Māori being in agreeance compared to non-Māori. The body condition score, which is strongly correlated with body fat mass, is based on the visual observation and palpation of superficial bony prominences and is a validated scale that classifies dogs into three groups: under-weight (1, 2 and 3); ideal weight (4 and 5), and overweight/obese (6, 7, 8 and 9) [41]. The latter group is associated with many health issues such as osteoarthritis, cardiovascular diseases, respiratory problems, and diabetes mellitus [29]. The results from this study indicate that being unfamiliar with the correct body condition associated with a healthy weight in dogs may be a potential risk factor contributing to pet dogs being overweight or obese. Lack of awareness of ideal body weight was identified as a possible obesity risk factor in a current study on Labrador Retrievers [42], and many studies have demonstrated that owners often misperceive their dog’s body condition [11,43,44,45]. Given that Gates et al. [16] recently reported a 28% prevalence of overweight/obese dogs in the North Island of NZ, it appears that the dog owners cannot be relied on to determine the appropriate body condition for their dog(s) and improved owner awareness needs to be a key priority for any healthy weight intervention for dogs in NZ.

### 4.3. Dog Feeding Practices

In this study, many survey respondents reported providing their dog(s) with specialised food (58%). However, only 28% of respondents indicated that they thought this should be sourced from a veterinary clinic or pet shop. Those respondents from households with children were less likely to consider a specialist source necessary. Some of the survey respondents also highlighted that good quality food could be purchased from supermarkets. Interestingly, Gates et al. [38] reported that the supermarket was the most common place for owners to purchase food (73% of dry and 92% of wet food) and that 19.4% of dog owners source food online, with a third of the online purchases being supermarket quality. However, some of the respondents raised concerns about supermarket dog foods regarding inadequate product labelling and packet information, claiming that this made it difficult to choose the appropriate food for their dog(s). Many overseas studies have investigated pet food safety concerns regarding inadequate information and/or mislabelling, with these being of particular concern for pets requiring a controlled diet [46,47,48,49]. In addition, significant variation in the mineral composition of dry and canned pet food in the United States of America has been reported by Paulelli et al. [50], with a significantly higher concentration of essential elements in canned foods compared to dry foods in the United States of America and dry dog food obtained from the supermarket in Brazil having iron concentrations above the maximum recommendation. Collectively these findings support the need for higher standards (nutritional and labelling) for supermarket dog food and suggest that supermarket-based body condition education would be well placed for addressing canine obesity in NZ.

Many respondents reported the feeding of specialised dog food in this study, but the food type was not disclosed. However, the bulk of the pet food being sold in any supermarket, pet store or veterinary clinic in NZ is dry. The popularity of dry food over other types has been attributed to its affordability, convenience, and long shelf life [9,51]. A number of studies have reported a positive association between obesity and consumption of dry food regardless of the dry food quality. It has been suggested that this is a consequence of dry food typically having higher calories per gram than wet food [9,14,52,53]. Other research has concluded that the overestimation of dry food could be a risk factor for weight gain, especially in small dogs, due to the increased inaccuracies of measuring smaller volumes of food [53,54]. Not surprisingly, dogs fed ad libitum (as often or as much as desired or necessary) have also been found to be at greater risk of weight gain [42,55]. Thus, the reported association of dog obesity and consumption of dry pet food may not only be related to calorie density but may also have been a consequence of overfeeding, with more food being consumed to reach satiety due to gastric extension [56,57]. Zicker [51] classified dog food into three different categories (dry, semi-moist, and canned) based on the water content. Approximately 28% and 20% of the respondents indicated that they feed their dogs dog rolls and wet food (canned), respectively. Lund and colleagues found that dogs fed semi-moist foods and canned foods had an increased risk of becoming overweight or obese [58]. In this study, dogs were also more likely to be provided dog roll if their owner was Māori, had a lower income, had a lower level of education, had more children, and/or were older-aged; some of these factors being indicative of a poorer socioeconomic status. The feeding of wet (canned) food was also associated with a lower income, indicating that dog roll and canned food may have been perceived as cheaper pet food alternatives. These results are perhaps not surprising given that low socioeconomic status (both at individual and neighborhood/area level) has often been associated with low diet quality and increases nutritional risk in humans [59,60,61].

A reoccurring theme that emerged from our data was that many owners provided their dog(s) with a varied diet and treats, nutritional enrichment being one way to enrich an animal’s environment (Bloomsmith et al. [62] classifies five types of environmental enrichment: (1) Social enrichment; (2) Occupational enrichment; (3) Physical enrichment; (4) Sensorial enrichment; (5) Nutritional enrichment). White and colleagues reported that 75% of dog owners consider treats an extra calorie intake rather than an integral part of their daily feeding intake [63]. Thus, a varied diet and feeding treats can make it difficult to control the caloric intake. This is reflected in Kour and colleagues’ findings, who identified that mixed diets and snacking were associated with obesity [42]. Similarly, Orsolya and colleagues reported that commercial dry food and/or leftover human food were associated with overweight dogs [28]. Likewise, other studies have found that feeding table scraps/human food are associated with overweight and obese dogs [63,64]. A third of the respondents in this study reported feeding their dog(s) table scraps, with many qualifying that it was occasionally as a treat and/or that only specific types of table scraps were provided. However, according to White et al. [63], leftover human food scraps are inappropriate for dogs, even as a treat, due to the high fat and salt contents [63]. Keeping in mind that an owner’s leftover and/or table scrap quality will solely depend on the quality of the owner’s diets, it is not surprising that some studies have reported an association between dog obesity and owner obesity [20,42,65].

While only a third of the respondents reported feeding their dog(s) table scraps/human food, more than half of the respondents reported feeding their dog(s) treats. However, the survey respondents broadly interpreted the term “treat” and treats were specified as being a wide range of things, such as ice blocks containing fruit and vegetables, table scraps/human food, dental chews, cheese, peanut butter, etc. and dog-specific. Many studies support the notion that the feeding of ‘treats’ is a potential obesity risk factor for dog/s [11,14,17,35,53,63,66]. In addition, some of these studies showed that increasing treat feeding frequency was associated with canine obesity [11,17,35,63,66], and others have suggested that owners of overweight/obese dogs show their affection to their animals by providing them with treats [35,67]. Interestingly, female respondents were more likely to feed their dog(s) treats, consistent with other studies that evaluated dog owners’ attitudes towards treats [39,63]. Our study also found that dogs were more likely to receive a treat if their owner was living in a city or town.

In contrast, it was found that dogs were less likely to receive a treat as the owner’s age and the number of children in the owner’s household increased. We speculate that the latter (non-gender related) observations could be due to accessibility and financial constraints. Further research is required to understand owner attitudes, beliefs, and behaviours around feeding treats to their dog(s) and how these impact dog obesity in NZ. However, with the growing awareness of the importance of environmental enrichment for improved canine quality of life [68,69], it will be important for the term “treat” to be clearly defined. In this study, the term was perceived to include everything from table scraps (obesogenic), through to training-specific treats and food enrichment activities (e.g., food puzzle games and food frozen in ice blocks), which enhance activity levels and therefore may be anti-obesogenic.

In contrast to the other food types in their study, Orsolya and colleagues reported that dogs fed raw food, either exclusively or in combination with other food types, were less likely to be overweight [28]. The anti-obesogenic association with raw food may potentially be due to the extra time needed to chew the raw materials and raw food being less calorie-dense than dry food. In the present study, raw meats were reported to be fed to dogs by 36% of respondents in the current study, with this feeding practice being more likely if the respondent had a rural upbringing and/or did not live in a town. These results agree with Companion Animal NZ’s findings [70], which reported that 37% of NZ owners feed their dog(s) raw meat, with this being more likely if their owners were in rural and regional areas. As the number of household adults increased, the likelihood of the household dog(s) being fed raw meat decreased. Interestingly, this was the only association with the number of adults in the household that was observed in this study. Given the anti-obesogenic association with raw food, our finding aligns with the findings of Bland et al. [11], who reported that obese dogs were more likely found in households with more adults. In NZ, supermarket meat is becoming more expensive [71], so again, we speculate that these observations reflect accessibility (with those not living in towns or cities having more access to home-killed meat) and financial constraints.

Some of the respondents in our study commented that the raw meat was part of a bone and raw food or biologically appropriate raw food (BARF) diet. However, the source, quality and quantity of raw meat being fed was not part of this investigation. Morgan et al. [72] reported that 24% of dogs’ owners in the United States of America that lived rurally fed their dog(s) raw meats, and the authors established that the quality and quantity of raw materials offered to the dogs by their owner depended on the knowledge and information gathered from online resources rather than veterinary clinics. Similarly, Connolly et al. [73] observed that breeders who exclusively provided home-prepared diets for their dogs gathered nutritional information from non-veterinary websites, books, or email groups that were considered more trustworthy, indicating a lack of trust in their veterinary clinics. Similarly, other studies have suggested that many owners consult websites to learn how to feed their pets [52,74]. Thus, while veterinarians remain the most trusted source of information [38,74,75,76] and are therefore well-placed to educate owners, credible non-veterinary web-based platforms are also important when developing a healthy weight intervention for dogs. In NZ, 21% of dog owners sourced their pets from breeder/hobbyist enthusiasts [1] that often provide extensive educational resources for owners and continued support and advice throughout the animal’s lifetime; thus, breeders and breed societies must have access to credible information and education.

Raw meat-based (RMBD) and BARF diets are gaining popularity with the growing perception that more natural unprocessed diets are healthier for pets [53,77,78]. Some studies have supported this view and suggested that compared to conventional and processed diets, raw meats have nutritional and behavioural advantages [53,77,78]. However, in contrast, several studies have suggested that raw diets, regardless of being commercial or homemade, may be nutritionally inappropriate for dogs and/or some life stages [79,80,81,82]. Furthermore, concerns have been raised about food preparation hygiene and potential pathogenic and zoonotic hazards [83]. Several studies have suggested that dogs fed raw meat may shed zoonotic pathogenic bacteria and parasites in the surrounding environment [74,84,85,86], posing a hazard to public health as well as to animals.

Additionally, some studies have reported that dogs fed raw meat bones are more likely to fracture their teeth, negatively impacting dental health [87,88]. Conversely, chewing raw bones has been reported to remove dental calculus, positively impacting dental health [89]. Thus, future studies to investigate and clarify the benefits and risks related to the raw feeding practice of dogs in NZ are required. Furthermore, easily accessible, high-quality information about how to correctly source, prepare, and provide raw food as part of a complete and balanced diet is essential if raw food is going to be promoted as an anti-obesogenic option.

In addition to the associations with food types discussed above, some studies have also found an association between obesity and the feeding frequency in dogs [10,90]. In this study, food type data was collected but other feeding protocol factors (frequency, duration, and quantity provided) were not included. Further research regarding the feeding protocol factors, activity levels, and the provision of environmental enrichment and their effects on the body condition, welfare, and wellbeing of NZ dogs is required. Such research should consider using validated questionnaires/tools such as the Dog Obesity Risk and Appetite (DORA) questionnaire [91] and a body condition score system [41], for example.

## 5. Conclusions

The current study highlighted several potential owner-related risk factors contributing to obesity in companion dogs in Aotearoa NZ. We have identified that owners are often unfamiliar or disagree with the ideal body condition for dogs. They also provided their dogs with a varied diet, including various forms of treats, making it challenging to track caloric intake. Future research that is representative of NZ dog owners is required to investigate the perception of body condition by both owners and the association of body condition with feeding protocols (type(s), frequency, quality, and quantity of food). Such research would inform educational programs that enable dog owners to more correctly identify canine body conditions and detect any changes, allowing owners to modify their feeding protocols accordingly. Additionally, owner education regarding the health risks associated with a dog being overweight or obese is also vital. Not surprisingly, we also found that social factors such as ethnicity, income, and the number of children also impacted the type of food given to dogs, with more obesogenic food options provided by respondents who were Māori, had a lower income, and/or had children. We therefore speculate that accessibility, time, and financial constraints drive these dog food selections and suggest that supermarket- and web-based platforms on healthy weight interventions (including education) from animal welfare and/or breed society perspectives (as opposed to having commercial links) are important due to their convenience and perceived trustworthiness. Regarding breeders and breed societies, systems must be put in place that ensure they are disseminating appropriate body condition and feeding protocol information. We also suggest that better food quality standards and easy-to-understand labelling will facilitate the maintenance of a healthy weight in dogs. Guidelines for providing dogs with different diet options depending on their optimal weight, nutritional value, dental health, and environmental enrichment are paramount and need to be developed using a pro-equity approach so that dogs in Aotearoa NZ live not only a good life but also their best life.

## Figures and Tables

**Table 1 animals-12-00267-t001:** Positive responses to the question “Which of these apply to your dog/s?” along with factors and variables that impact the likelihood * of a positive response in the EIT Pet Survey 2019.

What Do You Feed Your Dog/s?	*n*	Percentage	Associated Factor or Variable with Odds Ratio * (*p*-Value)
My dog is fed treats	880	59%	Gender: male/female 0.548 (0.011) Age range 0.917 (0.044) Town/city-living 1.341 (0.028) Number of children 0.767 (<0.001)
My dog is fed specialised dog food	862	58%	Household income 1.162 (0.005) Qualification level 1.089 (0.002) Number of children 0.739 (<0.001)
My dog is fed dog biscuits from the supermarket	583	39%	Household income 0.876 (0.014) Qualification level 1.904 (<0.001) Number of children 1.387 (<0.001)
My dog is fed raw meat	533	36%	Rural upbringing 1.439 (0.009) Town/city-living 0.714 (0.140) Number of adults 0.860 (0.037)
My dog is fed table scraps/human food	503	34%	Number of adults 1.269 (0.001) Number of Children 1.136 (0.046)
My dog is fed dog roll	414	28%	Ethnicity 0.564 (<0.001) Age range 1.118 (0.022) Household income 0.863 (0.014) Qualification level 0.890 (<0.001) Town/city-living 0.577 (<0.001) Number of children 1.188 (0.016)
My dog is fed wet (e.g., canned) food	299	20%	Household income 0.856 (0.011)
My dog is fed food that I have cooked for them	275	18%	Household income 0.847 (0.010) Number of children 0.789 (<0.001)

* Odds ratio >1 indicates an increased likelihood of a positive response, whereas an odds ratio <1 indicates a decreased likelihood of a positive response.

**Table 2 animals-12-00267-t002:** Thematic analysis of comments provided about respondents’ dog diets in the EIT Pet Survey 2019.

Category	Theme: Subthemes (If Applicable)	Representative Quote/s
Treats	For training	“Treats are very occasional/used for training”
	Occasional or rare	“Occasional treats = Apple slices, dog treats and small leftovers.”
	Different types: Ice blocks Fruit and vegetable Dental chews Animal specific	“We make carrot and cucumber icicles for him in summer and he sometimes get cheese and peanut butter as a treat” “The treats are dental chews, …” “She gets special dog treats”
Specialised food	Variety of sources: Online Breeder Specialist store (pet, veterinary, farm)	“He has more natural dry food without all the filler stuff added, purchased online”. “Food that comes from the breeder” “Puppy food from pet shop”, “Primarily vet kibble, but occasionally other items”. “Dry food from Farmlands”
	Health reasons: Recommended/prescription, Grain-free Allergies	“One dog is on a prescription veterinary diet.”, “One dog has a grain-free diet due to certain allergies” “We are very limited because she has severe food allergies”
	Used in combination with other types of food	“Currently she has some Hills prescription food mixed 50/50 with supermarket dog food.”
Supermarket dog biscuits	Good/high-quality biscuits	“I do not feed them the cheap biscuits from the supermarket. They get Purina”
	In combination with other types of food	“Additionally, certain kibble from supermarket and teeth cleaning kibble from the vet”
	Fussy (only eat supermarket biscuits)	“He is fussy and stopped eating the better quality biscuits I was buying him.”
Raw meat	Part of a raw food diet: Not just meat BARF	“Raw meat and a raw food diet are completely different. He is fed a raw food diet.”, “I follow BARF—bones and raw food”
	Specific type: Fresh Frozen Prey/game/roadkill	“Dry and raw lamb/chicken mince.” “Frozen free flow meat” “My dog is fed a balanced prey diet, mostly raw but not only ‘meat”, “I encourage eating roadkill (not squashed and rotting) possum, other game, fish.”
	Different sources: Bought commercially, Farm kill, Hunted	“Commercial raw dog food ” “Offcuts of our cattle beast” “My dog catches rabbits and possums”
	Bones: Types Health reasons	“Raw meat = beef bones and not actual meat” “About once a month he gets a raw bone and a rawhide bone to help his digestion and keep his teeth clean.”
	In combination with other types of food	“Mix of raw chicken, rice and other things”
	Occasional	“Occasionally given uncooked bones, raw meat and very occasionally treats or small amounts of table scraps”
Table scraps/human food	Selected scraps only	“Some table scraps depending on what it is”
	In combination with other types of food	“She eats both canned foods and people food. She loves broccoli and brussel sprouts”
	Occasional	“Fed human food very occasionally”
	As a teaser (on other food)	“We add a teaser to their dry food—egg or chicken or leftover meats/vege etc”
Dog roll	Type: Real meat/chunky, Good/High quality	“The dog roll is Harris meats made with very little filler & lots of real meat and fat.” “I buy a high-quality dog roll”
	In combination with other types of food	“Our dogs are fed dry food in the morning and a mix of dog roll, veges, meat etc at night”
	Training treat	“Dog roll only as training treats”
Wet/Canned food	As a teaser (on other food)	“…the odd scoop of canned to make it interesting”
	In combination with other types of food	“A bit of everything! They have biscuits and tinned food, but often have home-cooked food too. Especially meat and gravy.”
	Different types	“Wet food = tinned tripe”
Home-cooked pet food	Health reasons	“I cook for my fussy dog when she gets sick (quite often now that she’s old)…”
	Regular diet	“We cook up food …, we believe the processed food is not good for them at all.”
	In combination with other types of food	“Semi-homemade diet with air-dried dog food”
Other themes	Individualised diets (for dogs in the same household)	“Two dogs one old one a pup hence the variety of answers”
	Varied diet	“Fed a variety of foods, not the same every day”
	Fruit and Vegetables	“Additionally, fed appropriate vegetables and fruit”
	Cat food	“She eats cat biscuits from the supermarket”
	Supplements	“…, plus additional supplements if and when required”
	Dental routine	“Regular Dentastix as both treat and teeth care”
	Finances	“What I feed my dog is based on budget. I try to get him the best food that I can depending on how much money I have”

**Table 3 animals-12-00267-t003:** Percentage of 2019 NZ Pet Survey respondents selecting each level of agreement for the following statements regarding body condition and specialised pet food, respectively: “Dogs should have ribs, hips, and a spine that are not visible but are easily felt” and “Dogs should have a specialised diet from a pet shop or vet clinic”.

Statement about:	*n*	Strongly Agree	Agree	Total Agreed	Neutral	Disagree	Strongly Disagree	Total Disagreed
Body condition	2285	32.6%	46.0%	78.6%	16.2%	3.4%	1.8%	5.2%
Specialised diet	2287	8.1%	19.5%	27.6%	55.9%	13.4%	3.1%	16.5%

**Table 4 animals-12-00267-t004:** Thematic analysis of comments provided by the EIT Pet Survey 2019. respondents when asked how strongly they agreed or disagreed with the following body condition and specialised diet statements, respectively: “Dogs should have ribs, hips, and a spine that are not visible but are easily felt” and “Dogs should have a specialised diet from a pet shop or vet clinic”.

Statement about:	Theme: Subtheme (If Applicable)	Representative Quote/s
Body condition	Uncertainty: Breed differences Knowledge gap	“Animals should be of a good condition although some breeds are going to be more bony so the second to last question is difficult to answer.” “I am uncertain what a healthy weight for a dog looks like re feeling ribs etc”
	Animal differences: Breed Age Purposefully lean	“Re the hips and ribs question, the breed of the dog will depend on if you’re meant to be able to feel them. I.e. you can feel them on a boxer but you cannot on an English bulldog” “Our dog is 14 now so his spine and his hips are becoming more prominent with age.” “My dogs are all kept lean and fit for competition to ensure they do not harm themselves competing”, “For working dogs they are meant to be on the lean side otherwise struggle to work but house dogs ribs should not be felt”
	Disagree	“I do not think dogs ribs should be easily felt and could do with a little more weight on them than vets recommend but they should not be fat either”
	Individualised health focus	“A dog should be a healthy body weight and condition for its age, breed and lifestyle”
Specialised diet	Diet should be needs-specific (source not important)	“Does not have to be a specialised diet from vet but must be suitable for age/breed/type/behaviour and all other factors”
	As required: Recommended/prescribed Not always necessary	“Specialised diets should be provided if they’re needed, but not unless specifically recommended by a vet.” “Specialised diets are not always necessary—not all supermarket brands are terrible”
	For health reasons	“Our current dog is on a special diet due to health issues, but all our other dogs have just eaten commercial pet food from the supermarket plus table scraps, so I would only go down the special diet road if advised to do so by the vet”
Other themes	Better food standards needed	“I think supermarkets should stop selling awful pet food and the standard should be raised for what is acceptable”
	Financial influences	“If the dog is happy and healthy supermarket dog food is fine as that is all some people can afford”
	Other diet alternatives	“A really well educated owner might be able to feed a dog adequately on an alternative diet but would need to understand canine nutritional requirements.”

## Data Availability

The data presented in this study are available on request from the corresponding author.

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
