# Peer review of "Potential Owner-Related Risk Factors That May Contribute to Obesity in Companion Dogs in Aotearoa New Zealand"

_animals, 2022, doi:10.3390/ani12030267_

Round 1

Reviewer 1 Report

Well-written study on new Zealand owners attitudes towards feeding dogs and general attitudes on body condition. I think it is important to publish studies that include different areas of the world because the problem of dog obesity can certainly have a cultural component. I think a negative of this study is that they did not use a validated questionnaire (of which there are several). Also, they only looked at feeding, not exercise. I think the title is not really an accurate reflection of the data. The authors did not actually get any dog obesity measurements (either owner viewpoint or actual measurements). I think a more appropriate title is something like viewpoints of owners on feeding and body condition. I appreciate the body condition questions that they tried to use, but it sounds like they caused some confusion. Again, using a validated tool like the Purina 9 point scale might have cleared up confusion with the questions asked.  Even the feeding questions do not really reveal much about why dogs might be obese. Assuming that dry food feeding leads to obesity is not really entirely accurate.

Author Response

Comments and Suggestions for Authors

  • Well-written study on New Zealand owners’ attitudes towards feeding dogs and general attitudes on body condition. I think it is important to publish studies that include different areas of the world because the problem of dog obesity can certainly have a cultural component.

Thank you

  • I think a negative of this study is that they did not use a validated questionnaire (of which there are several). Also, they only looked at feeding, not exercise. I think the title is not really an accurate reflection of the data. The authors did not actually get any dog obesity measurements (either owner viewpoint or actual measurements
  • I think a more appropriate title is something like viewpoints of owners on feeding and body condition. I appreciate the body condition questions that they tried to use, but it sounds like they caused some confusion. Again, using a validated tool like the Purina 9-point scale might have cleared up confusion with the questions asked.
  • Even the feeding questions do not really reveal much about why dogs might be obese. Assuming that dry food feeding leads to obesity is not really entirely accurate.

The data used in this study was taken from a much larger survey that had a more general focus. Given the results of our exploratory study, we agree that more focused research is required around feeding protocols, activity levels and environmental enrichment and their associations with canine body condition and obesity, welfare and wellbeing. We have acknowledged this in the discussion and, in response to your comments, we have added that future research should consider using a validated questionnaire/tool (lines 500-507), and we have re-worded the title to reflect the study better.

Reviewer 2 Report

The manuscript is well written and presents interesting results obtained from one questionnaire, about the perception of owners about dogs body composition and about the role of the food in that. Moreover, the results provide evidences about factors associated with dog condition score, in NZ, providing relevant information that can be further used to promote dogs healthier food habits and body weight. 

Only the objective description can be improved, since in abstract it is more clear than at the end of introduction.

Apart from that, the manuscript is acceptable to be published.

Author Response

Thank you. We have re-worded the description of the objective to make it less wordy. “In this study, NZCAT research project data relating to the diets of dogs and the perceptions of body condition were explored to identify possible owner-related risk factors that may contribute to the development of canine obesity.”

Reviewer 3 Report

I read this story with interest and I have several critical (major) as well as minor comments

Minor:

  1. 41 – I am just curious what is top owned animal, because Europeans prefer dogs followed by cats
  2. 43: It has long been acknowledged that dog ownership is influenced by demographic variables such as the location and size of the house or property, the number of children in the family, education level and household income [2,3]. – here I would be more specific – is ownership always POSITIVELY influenced by the size of house (=larger house = greater likelihood of dog ownership), number of children (=more children = greater likelihood of dog ownership) education level (higher education level = greater likelihood of dog ownership – if the reverse was true, please change the wording) and income (higher income = greater likelihood of dog ownership)? I assume that yes, thus I would add the word ʺpositivelyʺ somewhere between ʺisʺ and ʺinfluenced. ʺ
  3. 71 – but the association between dog obesity/overweight and the owner obesity/overweight was significant in the cited paper (Linder, D. E., Santiago, S., & Halbreich, E. D. (2021). Is There a Correlation Between Dog Obesity and Human Obesity? Preliminary Findings of Overweight Status Among Dog Owners and Their Dogs. Frontiers in Veterinary Science, 736.)

End of Intro – given a number of research works in this field, i tis required to explicitly establish hypotheses here.

  1. 110 – 111 – my recommendation for the future is to ask exactly on age of participant and then you can make age categories. Actually it is impossible to calculate mean and SD or SE from these ordinal data.
  2. 339-340 – please refer to dog obesity
  3. 414 – 416 – I assume that people having less children, but caring more about dogs are those who are concerned with food from supermarkets. Can this issue be supported statistically?

Major:

  1. 146 Perhaps the frequency of dog feeding and dog health problems/frequency of visits in a veterinary could be asked.
  2. 158/9 – it is not clear to me how the binary statistics could be applied here if the owners could tick several possibilities regarding the dog diet. What exactly was 0 and what was 1? Please explain.

Recruitment of participants is not explained at all. Please provide detail how recruitment was realized, whether participants were paid or not, whether IP addresses were controlled or not etc.

  1. 200-216 – these characteristics are purely descriptive, not following any hypothesis, thus they could be presented in any table. I encourage the authors to re-think their strategy how to present results here.

ribs, hips, and a spine that are not visible – I am little bit confused with this statement. Consider you have a dog with thick coat (e.g. a retriever) against a dog with short coat (e.g. an English bulldog). The latter can have visible ribs, hips or spine, but it is much less visible in the former group of dogs. Can this be something like confounding factor in self-perceived dog overweight? Was this item also used in another works? If yes, please support this with reference(s). Moreover, ʺattitudesʺ in this study are not defined and, according to my view, they do not meet criteria for attitudes having behavioural, cognitive and emotional elements (c.f. Eagly, A. H., & Chaiken, S. (1993). The psychology of attitudes. Harcourt brace Jovanovich college publishers). Finally, attitudes are not investigated with only two items in psychological literature. I think that the authors refer rather to opinions, not to attitudes.      

  1. 296 – 312 – I recommend to use pictures or photographs of dogs of various condition (optimally the same dog overweighed, normal, underweighed) in your future research which can greatly helps to identify what each participant means/think.
  2. 437 – I assume dog fed ad libitum in a flat can has greater risk of obesity than a dog living in a garden of the family house, where movement constraints are smaller.

Conclusion: This study has highlighted several potential owner-related risk factors contributing 558 to obesity in companion dogs in Aotearoa NZ. – I think that proper estimate of dog condition needs to address following: 1) to provide dog breed and sex, 2) to provide its last body weight (exact weight should be separated from self-perceived and separately analysed statistically), 3. to provide a series of pictures showing underweight, normal and overweighed/obese dogs to help the participant to identify what is the body of his/her own dog(s). Consider e.g. research on perception of participant own body - the same approach can be adopted for perception of own dog(s).

It is a shame you did not investigate BMI of dog owners, perhaps you could easily contribute to the finding that owner weight correlates with its dog weight (Linder et al. 2021).

Overall, I perceive this study as an extension of existing literature, but it is hard to identify its novelty here. To my opinion, crucial estimates of dog body condition are not valid.  

Author Response

Comments and Suggestions for Authors

I read this story with interest and I have several critical (major) as well as minor comments

Thank you for your thorough review of our manuscript.

Minor:

  1. 41 – I am just curious what is top owned animal, because Europeans prefer dogs followed by cats

According to the Companion Animals NZ 2020 report, cats are the most popular companion animal in New Zealand, with 41% of households sharing their home with at least one cat. https://www.companionanimals.nz/publications

  1. 43: It has long been acknowledged that dog ownership is influenced by demographic variables such as the location and size of the house or property, the number of children in the family, education level and household income [2,3]. – here I would be more specific – is ownership always POSITIVELY influenced by the size of house (=larger house = greater likelihood of dog ownership), number of children (=more children = greater likelihood of dog ownership) education level (higher education level = greater likelihood of dog ownership – if the reverse was true, please change the wording) and income (higher income = greater likelihood of dog ownership)? I assume that yes, thus I would add the word ʺpositivelyʺ somewhere between ʺisʺ and ʺinfluenced. ʺ

The wording has been changed as suggested to “It has long been acknowledged that dog ownership is positively influenced by demographic variables such as a non-urban location, and increased size of the house or property, the number of children in the family, education level and household income [2,3].”

  1. 71 – but the association between dog obesity/overweight and the owner obesity/overweight was significant in the cited paper (Linder, D. E., Santiago, S., & Halbreich, E. D. (2021). Is There a Correlation Between Dog Obesity and Human Obesity? Preliminary Findings of Overweight Status Among Dog Owners and Their Dogs. Frontiers in Veterinary Science, 736.)

In this sentence, we were not referring to statistical associations. We have re-worded the sentence to improve its clarity. It now reads, “Although a similar obesity pattern has been reported in human and pet populations, the factors contributing to these associations have not been characterised [18-20].”

End of Intro – given a number of research works in this field, I is required to explicitly establish hypotheses here.

A hypothesis has been included towards the end of the introduction. “We hypothesise that the behaviours and perceptions of some NZ dog owners may reveal potential canine obesogenic risk factors.”

  1. 110 – 111 – my recommendation for the future is to ask exactly the age of the participant, and then you can make age categories. Actually, it is impossible to calculate mean and SD or SE from these ordinal data.

Thank you for your recommendation; we will take this into consideration when undertaking future research.

  1. 339-340 – please refer to dog obesity

We do not think it is appropriate to refer to dog obesity in our study as we did not collect obesity data on the animals. Therefore, direct associations between owner perception and obesity could be determined. Lines 341-356 make the potential link between unfamiliarity with or incorrect perception of body condition (the results of our study) and obesity.

  1. 414 – 416 – I assume that people having less children, but caring more about dogs are those who are concerned with food from supermarkets. Can this issue be supported statistically?

This notion is supported by the various Odds Ratios presented in Table 1, with decreasing number of children increasing the likelihood of being fed specialised dog food, whereas an increasing number of children increases the likelihood of being fed supermarket kibble and dog roll.

Major:

  1. 146 Perhaps the frequency of dog feeding and dog health problems/frequency of visits in a veterinary could be asked.

The data used in this study was taken from a much larger survey that had a more general focus. Given the results of our exploratory study, we agree that more focused research is required around feeding protocols, activity levels and environmental enrichment and their associations with canine body condition and obesity, welfare and wellbeing. We have acknowledged this in the discussion and, in response to your comments, we have added that future research should consider using a validated questionnaire (lines 500-507).

  1. 158/9 – it is not clear to me how the binary statistics could be applied here if the owners could tick several possibilities regarding the dog diet. What exactly was 0 and what was 1? Please explain.

Each dog food type was assessed individually, with the binary aspect being did the owner select this food type (yes = 1, no = 0). This has been clarified in section 2.2.

  1. Recruitment of participants is not explained at all. Please provide detail how recruitment was realised, whether participants were paid or not, whether IP addresses were controlled or not etc.

As mentioned in the first paragraph of the Materials and Methods, a full description of all aspects of the research can be found in the Furry whānau wellbeing: Working with local communities for positive pet welfare outcomes report [23]. We have provided more detail in section 2.1. “The survey was promoted through various social media (for example, Facebook, Instagram, LinkedIn) and via email using a snowballing approach. Participants were not paid, and IP addresses were not controlled for as it was expected that some households might share devices.”

  1. 200-216 – these characteristics are purely descriptive, not following any hypothesis; thus they could be presented in any table. I encourage the authors to re-think their strategy how to present results here.

Ribs, hips, and a spine that are not visible – I am little bit confused with this statement. Consider you have a dog with thick coat (e.g. a retriever) against a dog with short coat (e.g. an English bulldog). The latter can have visible ribs, hips or spine, but it is much less visible in the former group of dogs. Can this be something like confounding factor in self-perceived dog overweight? Was this item also used in another works? If yes, please support this with reference(s).

This description is based on the description provided for the1-9 point body scoring system in which the ribs, hips and spine are visible in underweight animals but are not all visible in an animal of ideal weight though all can be easily felt, whereas, in an overweight animal, not all can be easily felt.

Laflamme, D. Development and Validation of a Body Condition Score System for Dogs,” Canine Practice, Vol. 22, No. 1, 1997, pp. 10-15

The issues raised here about confusion are discussed in lines 254-270

Moreover, ʺattitudesʺ in this study are not defined and, according to my view, they do not meet criteria for attitudes having behavioural, cognitive and emotional elements (c.f. Eagly, A. H., & Chaiken, S. (1993). The psychology of attitudes. Harcourt brace Jovanovich college publishers). Finally, attitudes are not investigated with only two items in psychological literature. I think that the authors refer rather to opinions, not to attitudes.      

This opinion contrasts that of reviewer 1. Therefore, we will leave this to the editor's discretion and amend the word attitudes in the manuscript to opinions if directed to do so. Furthermore, the word attitude is not in the title or abstract. We have removed the word attitude from the keyword list and any headings in the manuscript.

  1. 296 – 312 – I recommend to use pictures or photographs of dogs of various condition (optimally the same dog overweighed, normal, underweighed) in your future research which can greatly helps to identify what each participant means/think.
  2. 437 – I assume dog fed ad libitum in a flat can has greater risk of obesity than a dog living in a garden of the family house, where movement constraints are smaller.

These aspects were outside of the scope of our exploratory study but your comment will help inform our subsequent research.

Conclusion: This study has highlighted several potential owner-related risk factors contributing 558 to obesity in companion dogs in Aotearoa NZ. – I think that proper estimate of dog condition needs to address following: 1) to provide dog breed and sex, 2) to provide its last body weight (exact weight should be separated from self-perceived and separately analysed statistically), 3. to provide a series of pictures showing underweight, normal and overweighed/obese dogs to help the participant to identify what is the body of his/her own dog(s). Consider e.g. research on perception of participant own body - the same approach can be adopted for perception of own dog(s). It is a shame you did not investigate BMI of dog owners, perhaps you could easily contribute to the finding that owner weight correlates with its dog weight (Linder et al. 2021).

As mentioned above, the data used in this study was taken from a much larger survey that had a more general focus. Given the results of our exploratory study, we agree that more focused research is required taking the aspects you have mentioned into consideration. However, to gain funding to do such research we first had to ascertain that it was necessary for the NZ context. Our results will facilitate future research in these areas.

Overall, I perceive this study as an extension of existing literature, but it is hard to identify its novelty here. To my opinion, crucial estimates of dog body condition are not valid.  

Thanks for your comments, but your opinion contrasts Reviewers 1 and 2.

Round 2

Reviewer 1 Report

The changes are well done

Reviewer 3 Report

Thank you for your replies and improvements. I do not think that my opinion contrasts Ref 1 and 2. The review of Ref. 1 cannot be objectively evaluated, because he/she has almost no comments. Ref. 2 clearly recognized that dog obesity has not been adequately assessed, thus I see rather agreement than disagreement.